# Quality of Work Life and Work Process of Assistance Nurses

**DOI:** 10.3390/ijerph19116415

**Published:** 2022-05-25

**Authors:** Denisse Parra-Giordano, Denisse Quijada Sánchez, Patricia Grau Mascayano, Daniela Pinto-Galleguillos

**Affiliations:** 1Departamento de Enfermería, Universidad de Chile, Av. Independencia, Santiago 7760388, Chile; dquijada@uchile.cl (D.Q.S.); patricia.grau@uchile.cl (P.G.M.); 2Programa de Microbiología y Micología, Facultad de Medicina, Universidad de Chile, Av. Independencia, Santiago 7760388, Chile; dpintogalleguillos@gmail.com

**Keywords:** occupational health, quality of life, nurses, working conditions, work, primary health care

## Abstract

Background: The concept of Quality of Work Life (QWL) has been built multidimensionally through social reproduction; it is impacted by the perceptions of each individual and by the relationship between workers and the work environment. Objective: to analyze the Work Process and QWL of assisting nurses in public health. Methods: Research in a critical paradigm, descriptive, exploratory with a qualitative approach. The population corresponds to nurses who work in care work. Semi-structured guiding questions were applied and were analyzed with content analysis. Results: seven participants declared female; all Chilean; seven are young adults; six singles; only one has children, and one has a person dependent on her care; six are heads of household, and five receive help with housework. All have a nursing degree, five have a diploma, but none have a postgraduate. The work process has three subcategories: work object, instrument, organization, and work conditions; the QWL category has six subcategories: definition and perception of QWL, QWL potentiating factors, QWL exhausting factors, QWL improvement strategies, the emotional burden associated with QWL, and health problems. Conclusions: In this way, the lifestyle built by the assistance in the health area has repercussions on the quality of life and health in general.

## 1. Introduction

Work is a central aspect of life, being a means to obtain social and economic benefits and position individuals in society, determining their acquisitive power [1].

The concept of Quality of Work Life (QWL) has been multidimensionally constructed through social reproduction, which determines their social insertion historically, considering the work performed by each person [2]. QWL is promoted by favorable working conditions [3], impacted by the perceptions and feelings of each individual [4], and by the relationship between workers and the work environment [5]. The QWL is also understood as the worker’s satisfaction with their personal and work needs in the work process in a given position [6]. Must be valued and respected by the employer [4].

Nursing is the largest workforce in the health care system [5,7]. Therefore, QWL is essential for the nurse, recognized as the degree to which their personal needs are satisfied through quality care and organizational objectives [7,8]. However, many factors influence it [5]. Likewise, the nursing QWL plays a fundamental role in managing care [9], increases productivity, and reduces turnover and psychosocial impact [6,8].

Although it is thought that nurses, having the necessary knowledge and means, should provide an optimal QWL, different problems arise in their work development [10]. Likewise, globalization has brought changes in employment conditions and nursing work processes under the axis of capitalism, which has implied transformations in the work profile, and workers to adapt to the current economic model [11].

Nursing evolved from a trade to a professional career, with the work of Florence Nightingale being decisive in the formation of modern nursing. Under this model, nursing develops in five work processes: assistance, management, teaching, research, and political participation [11]. According to this classification, in Chile, assistance nurses work directly with people, providing care at different levels of care.

The work process has three elements: the work object where the activity is carried out; the instruments that are a set of tools between the worker and the object; the action that refers to work, its specific organization for a particular purpose and the conditions in which it is carried out [1]. Thus, for example, the healthcare nurse transforms the object (subject) using specific knowledge and own instruments.

The QWL of people is a historical and social result. However, on the other hand, the QWL of a group is determined by the time of life and work of everyone in society. Therefore, in terms of her work, each person develops a contradictory system of opposing forces of favorable and destructive processes that impact their Quality of Life (QoL) [11].

In this context, nurses’ work situation is impacted by how health institutions are inserted into the labor market, influencing working conditions, explicitly nursing as a social practice. Regarding the working conditions of nurses, the Labor Code regulates the legal processes [12], and the Health Code [13] establishes the areas covered by article 113 “includes the management of care concerning promotion, maintenance, and restoration of health, prevention of diseases and injuries”.

Professional burnout is a syndrome that has been the subject of many investigations, impacting their work and professional lives [14,15]. For example, direct care work is recognized as a source of occupational risk [14], even more so as nursing closest to the sick person [15]. Describing dissatisfaction with work, physical and emotional exhaustion [15], likewise, interference in family life.

The neoliberal economic model impacts the work of nursing; the consumer society encourages workers to have new houses, cars, and others. The National Research on Employment, Labor, Health, and Quality of Life of workers in Chile (2009–2010), refers that the worst evaluated aspects are the amount of money that enters the home and the level of debt [16]. In this way, healthcare nurses look for work sources outside the working day and in other institutions, taking time for family dedication and rest that impacts QWL.

Like any work process, care nursing generates favorable and destructive factors [14,15]. Therefore, the care nurse must know how working conditions affect the QoL and improve the quality of care delivered to the beneficiary population in the health network. Likewise, this knowledge will also promote health at work; these are direct contributions to the development of nursing in its practical and social context. Regarding the development of care, in Chile, a professional can develop in the three maintenance levels of the health network [17]. Furthermore, most studies are produced at the tertiary level, making it essential to carry out an approximation at the primary level.

Thus, because of the already known impact of the labor market on nursing professionals worldwide, it is necessary to broaden the knowledge of the nursing role that works in care in Chile. Therefore, this article aims to expand the lowdown on this topic, setting out the objective of analyzing the work process and the quality of work life referred to by public health care nurses.

## 2. Materials and Methods

Research in a critical paradigm, descriptive, exploratory with a qualitative approach.

Carried out in two Family Health Centers Primary level of care and a Type four Hospital of Santiago de Chile’s Western Metropolitan Health Service. The population corresponds to nurses and nurses who work in care work. Inclusion criteria: workday 22 a 44 h (half or full workday), care exercise for at least six months in the workplace. Exclusion criteria: professionals on vacation or medical leave.

Instrument: According to the literature review, the researchers created the question guide for this research based on two previous categories, work process and quality of work life. Then, they applied the guiding questions to nurses with the same characteristics to adjust their relevance to the object of study. Thus, the researchers asked sociodemographic characterization questions and two semi-structured guiding questions: (1) What is your work process at the primary level in public health services? (2) What is your perception of the quality of working life at the primary level in public health services?

Data collection: The researchers contacted the head nurse of each institution to coordinate the interviews, inviting all nurses in each workplace to participate. According to time availability, individual interviews were agreed upon, developed by the researcher without a prior relationship with participants. After explaining and signing the informed consent, five interviews were conducted at the Primary Health Care (PHC) of care and three at the tertiary level until the object of study was saturated. The participants agreed to participate in the investigation (there were no refusals or desist from the interviews) in a room of each institution with total privacy. With 17 to 37 min, from April 2019 to December 2020. They were recorded and transcribed to guide the analysis.

Data analysis: The researchers analyzed the sociodemographic data with descriptive statistics (absolute and cumulative frequencies) and the interviews with content analysis methodology according to Bardin [18], considering the predefined and emerging categories. According to the analysis developed from the interviews, they obtained three subcategories from the previous work process categories: work object, instrument, organization, and work conditions; the quality of work life category is made up of five subcategories: definition and perception of QWL, QWL potentiating factors, exhausting factors of QWL, QWL improvement strategies and emotional burden associated with QWL. Likewise, the emerging category of Health problems emerges. The investigators identified each interview with the letter E, according to the consecutive number.

Aspects of rigor and ethics: the rigor criteria of the study, according to Guba [19], were applied.

## 3. Results

This research had eight participants (eight interviews). The sociodemographic characterization is in Table 1.

### 3.1. Category 1. Work Process

The work process is built by the object, instruments, organization, and working conditions. Next, each subcategory is described with a summary result, and then add the individual respondents’ statements to support it.

#### 3.1.1. Subcategory 1.1—Work Object

The Work object is defined as gratification and pride for work, commitment, and vocation with the care of people.

*“For my nursing, it is the maximum in human terms; everything is very important, very enriching. I feel that not all people can work in what I work and it gives me a certain pride to be a nurse.”* E3

*“For me, my goal is to give a good quality of care to the person.”* E7

#### 3.1.2. Subcategory 1.2—Work Instruments

The Work instruments are determined from the institution or be personal. *“What I need is a box first of all, because if not, I can’t. Computer, I use my cell phone a lot; it’s a great tool that helps me a lot because I’m connected to the mail all the time, which drives me. And schedules, notebook ”*. E3

*“The cell phone with the Internet, but sometimes I also walk with my notes.”* E7

#### 3.1.3. Subcategory 1.3—Organization and Division of Work

The organization and division work are identified: working hours, distribution of tasks, salary, environment, request for permits and perception of medical licenses, support for training and well-being, labor relations.

Six have a day shift, and two have a shift system; work seniority ranges from 1 to 23 years, with 12 to 1 years at the institution. Regarding the development of their work, three are sector nurses; two are in charge of the transversal program, one is in order of the shift, and two are clinical floor nurses. According to the commute to work, the minimum time is 15 min, and the maximum is 90 min, four use public transport, three own cars, and one owns a bicycle. Salaries also present a variability from $950 and $1650.

*“The director chooses the positions. The director is the one who appoints you, and you are in charge of several tasks. Mostly, the unit boss manages the plan and my schedule. He is the one who sees your plans, blockades, your administrative hours, the hours that you see patients, etc.”* E3

*“In other words, I stay here, I go home, and I continue working on things that were pending. I have to review the files, and I have to see this other thing, but I never go on time ever.”* E4

*“I find work-relations good in general and, like in most places, there are always frictions within some people due to their personalities, I think, but in general, it is good. I don’t know, but if I start to compare other places’ stories, I find that the atmosphere here is good.”* E5

*“I have a fixed shift; my duties are defined in their protocol and everything.”* E7

### 3.2. Category 2: Quality of Work Life—QWL

The QWL is formed based on the definition and perception, potentiating factors, exhausting factors, and emotional charge.

#### 3.2.1. Subcategory 2.1—Definition and Perception of QWL

The QWL is defined and perceived as each person’s expectation, feeling full at work and having personal time.

*“Quality of life as the ideal that all the people who work should have, not being stressed, not worrying, not fighting, not arguing.”* E1

*“Quality of life I believe that when one feels as good, as full in what one does, what he is doing. If he is with the individual part well, I think that is what it is, quality of life. I believe that it is not pure to work, but to have a little of everything, that is, to have time with the family, go out, have fun, for everything, and work too.”* E4

*“I see work as very enriching for me, so for me, my quality of life at this time that has occurred at work is super good.”* E8

#### 3.2.2. Subcategory 2.2—Potentiating Factors QWL

The potentiating factors QWL are a favorable work environment, having the necessary resources to develop work, closeness to the home, good relationship with colleagues, consideration in decision-making, and assessment at work.

*“Yes, first of all, the work environment is favorable, I find it to be good, the conditions. I feel that this building is wonderful, everything is very clean. I have all the necessary resources to work well; I have nothing to complain about.”* E5

*“I think that for me at this moment it has a positive influence, one that is close to work where I live, that I feel comfortable where I work, that my peers are cooperative and we all jump up to finish a shift well, that the boss takes into consideration what I think and encourages that, that one feels comfortable and valued in the place where he works, I think that is why I would not change my job.”* E8

#### 3.2.3. Subcategory 2.3—Exhausting Factors (Negatively Affect) the QWL

The exhausting factors (negatively affect) the QWL are built by a long working day, the lack of resources that must be paid for personally, personal relationships, and the arrival of new professionals with a lack of preparation.

*“External things, inputs could also be that they could facilitate our work” “then I have to pay for the supplies because no one was going to pay for it here, so they are things like that, as they should worry a little more about quality with that one works.”* E1

*“I think personal relationships have a lot of weight, that is, relationships with your peers, with the heads, especially with the heads.”* E3

*“Just like the colleagues too, suddenly they come very unprepared, I feel that there is not good training now, that’s why I told you how the university is because I think that the colleagues, from other universities, do not come very well prepared; then it seems to me, that I think it is not favorable, there is no well-defined selection.”* E4

*“I consider that my work is exhausting because it is twelve long hours of work.”* E8

#### 3.2.4. Subcategory 2.4—QWL Improvement Strategies

The QWL improvement strategies are formed from activities to channel emotions, knowledge among team members, self-care activities; they can be from the company or personal.

*“Do a workshop, go to an activity that helps us to channel our emotions a little.”* E1

*“Being able to get to know each other more because one does not know each other here, he is known for the work we do, but one does not know either what problem the other colleague may be carrying.”* E2

*“What happens is that some time ago we had self-care activities, we had healthy breaks, we had a person who came to do gymnastics, they gave us massages, things, as well as every so often, we had something that took us out of the work thing, but now like one or two years ago we have been without that.”* E4

#### 3.2.5. Subcategory 2.5—The Emotional Burden Associated with QWL

The emotional burden associated with QWL is built from the development of the function of prioritizing care, receiving criticism from other professionals or relatives for lack of resources, and poor treatment of patients when things are not done personally.

*“Or sometimes you can’t cope with seeing all of them. Then you have to do a TRIAGE on who is worse, who comes later then maybe that is a burden as well as vital because it depends on you that the doctor sees that patient on the day Today, or why don’t you see him first and not this one, then there are times when the patients are both super serious. You don’t know who to prioritize. That is sometimes a bit harder.”* E1

*“I don’t care if I don’t have a box because I still come happy and I don’t care, I solve it as I can, but for me, if I got on severely with my class, with all the nurses, even if I had a box and everything that they gave me, it would not be so, that is, it would be more difficult.”* E3

*“Some doctors or some relatives do not understand that we lack things to be able to do some things or follow-up, so I did not walk with my head one hundred percent as to be able to withstand so much pressure within everything. Because the family members are the same here, doctors treat people badly when things are not done their way.”* E6

### 3.3. Category 3—Health Problems Associated with Work

It refers to health problems identified with an occupational cause by the interviewees. They are making up for difficulties falling asleep due to work problems, depression, anxiety disorder, workplace bullying, health problems associated with work stress.

*“Sometimes it is a bit difficult for me to fall asleep, thinking about some work issues, as well as about a patient, how will the lady be so much that she was about to die, things like that you turn around or small things like tomorrow I have to do this, thinking about what I have to do tomorrow and what I did not do and what I have to do.”* E1

*“Yes, but no, feeling depressed, feeling bad, leading me to go to a psychologist or psychiatrist for a work issue or stress. I have not felt that way yet, and I am not too young for that, I think.”* E2

*“I was with a depression, a depression together with an anxiety disorder that was produced by an occupational disease, by a workplace harassment, I was 3 to 4 months on leave, with psychiatric treatment.”* E3

*“Yes, last year I had a TIA, an accident because I was very stressed that time; I kind of stressed a lot because besides I had a lot of trouble, I was on an emergency shift.”* E4

*“Yes, or of course, probably because of the stress as well.”* E5

*“We get quite stressed by the lack of supplies.”* E6

## 4. Discussion

### 4.1. Sociodemographic and Labor Characteristics

As in the present research, in various studies on nursing care work the majority were women, Mexico 95.7% [20], Saudi Arabia 95% [21], China 94% [9], Malaysia 93% [22], Tehran Iran 90% [23], Kashan Iran 83% [24], Brasil 83% [25], Chile 83% [26], Irak 73% [4], Libano 60% [27]. In contrast, in Ethiopia, 51% were men [8]. Additionally, in most studies the participants have an average age of young adults (20 to 39 years) [4,5,8,9,20,21,22,24], Copiapó Chile [28]. Conversely, average age is 41 years [26]. Concerning marital status the marriage prevails in most studies [4,9,21,22,25,27,28], 68% had children [25], and 52% had responsibilities for the care of the elderly [9]. In other studies also the majority were graduates [4,21,23,27,28], but if they present diploma degree 96% [22], 87% had a higher degree [9].

### 4.2. Work Process

In other countries, different results are found in this research, where most of the workers are without a contract [4,20,26], only half had a full-time job, the most worked with a shift system [25,26,27], and they worked more than the contracted hours [8,25]. On the contrary to the present study, other research presents an average monthly income of CLP 500 [4], between CLP 1500 to CLP 3000 [26], more than CLP 1000 [28]. Complementing, most workers did not receive a risk allowance or certification compensation [8].

### 4.3. Definition and Perception of QWL

One study shows a high level of QoL, and the highest values are in managerial support and workload [28]; besides, the majority of nurses report being satisfied with the QWL in terms of the organizational culture [5]; and the well-being is higher in women [9]. Other research presents nursing QWL is mainly at a moderate level and needs improvement interventions [7,29], and the QWL is in the middle range [9]. On the contrary, more than half of the nurses reported a poor QWL [4,8]. This background implies a greater probability of leaving their job due to dissatisfaction with most of the components of the QWL (adequate and fair pay, safe and hygienic work environment, general atmosphere of work-life and social dependency of work-life) [4]. As evidenced in the present investigation one of the factors that influence QWL in the Nurse is stress [30]. Contact with death, patients and their families, conflicts with supervisors, and uncertainty about the therapeutic effect significantly increased pressure among participants [31]. Physical health has the best QoL and the lowest environmental field, mainly due to occupational stress [27].

Most PHC nurses are dissatisfied with work and family life factors, spend long hours, and are often exhausted after work [6]. In this way, nursing is perceived as strenuous work [31]. In Mexico, a study showed that dissatisfaction could be found in the QoL of the nursing professional in public health clinics in all dimensions. The dimensions with the most significant blow were inclusion in the workplace, personal development, and well-being achieved through work. The dimensions with the minor dissatisfaction were institutional support, free time management, job satisfaction, and job security [20]. In Chile, the perception of quality of care is acceptable, and the working environment conditions are evaluated as good except for the temperature [26].

### 4.4. QWL Potentiating Factors

As mentioned in the present research, organizational commitment negatively influences nurses’ QWL and intention to change [22]. As mentioned in the narratives, managers can do self-care workshops or other strategies to improve the QWL by providing opportunities for growth and development of staff skills, providing conditions for greater employee engagement, promoting teamwork, and planning fair and equitable pay [23]. The nurses acknowledge that they have received cooperation from their respective departments, supporting their QWL [5]. The higher quality of professional life, the greater job satisfaction, presenting a mutual and strengthening relationship between the two [28]. QWL the highest scores were for job and professional satisfaction [9]. Some studies show that the QWL of nurses has significant positive associations with: education, marital status, work experience, night shifts, monthly per capita income, support from family members [5]; years of experience (1–3), number of night tasks (5–7 days), number of extraordinary tasks (3–4 days), work on days off, rest at the right time [7]; organizational commitment, positive work environment, and satisfaction of staff [23]. Likewise, the public image of a society, the nursing position, the department of work, educational level, availability of a safe place to rest, availability of drinking water, the factors of age, title, education, employment contract, sleep disorder, and nursing adverse events were associated with a good nursing QWL [8,9].

Regarding the position or training of assistance nurses, the QWL of the head nurse is three times better than that of the rest. Furthermore, the QWL of the nurses’ undergraduate level is 2.3 times more than those who only have a diploma. It could influence the present investigation that there are no bosses or postgraduate degrees. In addition, the public image of a society, labor department, availability of a safe place to rest, and the availability of drinking water are associated with good QWL [4], and the socio-professional relationships [25] the latter as well as the present investigation.

### 4.5. Exhausting Factors of the QWL

As the nurses refer, nursing performance conditions are adverse, which may negatively impact their QWL and staff and their care. They felt helpless knowing the poorly performing care, expressing frustration and intention to change jobs and professions [32]. Complementing the stressing factors mentioned in this research in decreasing order, they are work overload, lack of supplies or poor quality of these, unsatisfactory salaries, number of nursing professionals inadequate to the demand, little clarity in functions, also mentioned as role ambiguity or multitasking, lack of security, lack of recognition, lack of opportunity for promotion, training and or updating, lack of benefits and rewards, lack of autonomy and participation, and poor organization [32]. Likewise, education, job title, workplace, a second nursing job, and age [27,28], time demand, frustration, effort levels, working rotating are significantly correlated with low QWL levels [24,27] are predictors of low QWL [24]. It is also essential to mention that the factors that cause dissatisfaction in nurses are inclusion in the workplace, personal development, and well-being achieved through work [20], communication difficulties, access to information to perform tasks, lack of support for professional development, lack of recognition, lousy work environment, and overload; besides, the last three were associated with low back pain [25]. The research participants mention occupational stress, a significant predictor of low QWL for nurses [9,27,31], while also influencing patient outcomes [31]. In addition, most studies affirm that the relationship between adverse working conditions and nursing health generates stress and job dissatisfaction. Body pain in the back, lower extremities, migraines, poor performance, discouragement, irritability, and depression [32]. In complement, a study in Bangladesh emerged seven barriers to the nurses working life: heavy workloads; lack of government accommodation and transportation; poor health; lack of support from nursing supervisors; lack of promotional opportunities; incomplete hospital policies and procedures; and lack of night shift and risk allowances [33]. For example, almost half of nurses wanted to quit their job in Iraq due to these conditions [4]. Therefore, improving QWL is essential to attracting and retaining employees, especially in healthcare [24].

### 4.6. QWL Improvement Strategies

As mentioned in the narratives of this article, in the first place, administrators must adopt effective strategies to propose favorable working conditions, consider work stressors, promote a better QWL in nursing workers, and optimize their work environments [4,8,27,33]. In addition, the use of preventive and efficient practices to manage stress seems essential [30], incorporating the levels of commitment to your organization [22], because the organization plays a critical role in determining the health risks associated with burnout syndrome [28].

As previously referred maintaining a healthy work-life is crucial for nurses to improve their QWL, increase retention, improve performance and productivity, promote safe nursing care [6], strengthen the educational level, and their awareness of community perceptions of nursing [8]. QWL improvement interventions include long-term care team talks, empowerment models, integration of focus groups and brainstorming roundtables, positive thinking training, stress inoculation programs, and participatory teamwork [29]; a stress inoculation program proved to be a cheap, safe, and effective way to improve the QWL of nurses [30].

The PHC research participants equally and other studies presents important pressure factors for nursing; besides rotation, they must consider developing improvement strategies in the QWL [6]. Interventions to improve the organizational climate will prevent job desertion by directly affecting the QWL; that is, a high QWL will retain the organization’s employees [4]. Hospital managers can reform the environment, the programming, and the system according to the different factors that affect the QWL of the nurses, carry out the psychological evaluation, and the appropriate behavioral psychological intervention for the nurses to improve the QoL [9].

A greater understanding of the nursing work process and its relationship with nonspecific low back pain pointed out that changes must occur in the organization and working conditions to reduce the risks of nursing workers becoming ill [25]. In addition, it is necessary to pay more attention to the low QWL and its related factors, significantly the workload, to understand that a high QWL positively influences the quality of care [7,20,24,33].

### 4.7. The Emotional Burden Associated with QWL

Because of the multiple stressors that assistance nurses are faced with, as reported in this research, they report the highest frequency of stressful events related to their healthcare team workload [27]. So too, several variables can significantly influence QWL, age, marital status, and education, critical in developing programs to improve the quality of work-life of nursing [21]. Regarding psychosocial work conditions, the literature also refers that stress is one factor that has the most significant negative impact. However, it also mentions the type of contract that is not named a problem in this article [26]. Being one of the main concerns precisely the contractual one to guarantee job stability and continuity of labor rights [26]. Additionally, workload and dealing with death are the most common work sources for nurses, resulting in the highest frequency of occupational stressors [27,28].

### 4.8. Health Problems

While in Chile, almost half of the nursing professionals perceive their health condition as very good. The other half is as good [26], as in this investigation, nurses describe many problems and barriers that affect their working life, feel the little reward for their hard work, and their health suffers from their working conditions leading to a low QWL [33]. There is also a relationship between work organization and working conditions with low back pain, meaning moderate risks to occupational disease. Likewise, factors that cause job dissatisfaction, such as lack of recognition, lousy work environment, and overload, are associated with low back pain. In addition, those who say they are moody, fatigued, and overloaded have low back pain, which increases their low back pain. Regarding nutritional status, just over half are above the normal BMI, and less than half are normal weight [25]. The main limitation of this research is its conduct only from the public perspective (both primary care level and hospital level); the integration of private institutions for further research should be considered, complementing the current one. Likewise, having started the interviews at the primary level, only three interviews were conducted at the tertiary level of care.

## 5. Conclusions

The development of this study allowed us to fulfill the proposed objective.

The main results referred to the work process and QWL describe in six subcategories: definition and perception of QWL, QWL potentiating factors, QWL exhausting factors, QWL improvement strategies, the emotional burden associated with QWL (previous categories), and health problems (emergent category).

In this way, the lifestyle built by assistance in the health area affects the quality of work-life and health in general. Thus, the present study is an innovative contribution to the development of nursing by making visible how working conditions affect the QWL of healthcare nurses and unveiling the healthcare network’s healthcare work process. Considering that nursing care work compromises the QoL of these professionals, the scarcity of studies on QWL in the nursing care network in Chile, it is expected that this study’s contributions will be taken as input to deepen the subject.

## Figures and Tables

**Table 1 ijerph-19-06415-t001:** Sociodemographic characterization (*n* = 8) Santiago, Chile (2019–2020).

Characteristics	Distribution in the Sample
Gender	
Female	7
Masculine	1
Age (average)	35 years old
Young adults	7
Average adult	1
Nationality	
Chilean	8
Marital status	
Single	1
Married	1
As a couple	6
Professional training	
Nursing degree (different universities)	8
Have a diploma	5
Have a postgraduate degree	0
Others	
Have children.	1
One person dependent on her care	1
Heads of household	6
Receive help with housework	5

## Data Availability

Authors have available the recordings of the interviews and all the transcripts of each one of them just as they have the analysis in its entirety.

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
