# Peer review of "Quality of Work Life and Work Process of Assistance Nurses"

_ijerph, 2022, doi:10.3390/ijerph19116415_

Round 1

Reviewer 1 Report

  1. According to the title of the article, the subject of this study was nurse assistants, but the role of the subjects cannot be found from the methods. The nursing system of each country is different. The author should add information related to the classification of nursing staff in their country and make it clear that these subjects are assistant nurses.
  2. The subject of the article is different from the keywords. For example, "occupational health" only appears in the keywords; "quality of work life" and "quality of life" have different meanings; "work process" does not appear, instead "working conditions".
  3. In qualitative research, the sample was not representative because it was small and non-random. Too many statements of personal characteristics in the article were unnecessary. The personal characteristics of the respondents in the abstract, discussion, and conclusion should be simplified.
  4. There were 8 respondents in this article, but the authors only cited a few of the descriptions in the results, and did not make a generalization of the results of all the respondents. This method of presentation was not general and would cause bias. It is suggested that authors should describe the summary of all the respondents’ answers in each paragraph, and then use individual respondent’s statements as reinforcement.
  5. Many literatures were cited in the discussion, but no literature was compared with the results of this study. Therefore, it is not known whether the results of this study are valuable, nor can the results of this study inspire them.
  6. The conclusion should state the important findings of the results of this study. It is not necessary to state the personal characteristics of the subjects, and there is no need to repeat the objective of the research.

Author Response

First of all, dear reviewer, I appreciate your time and comments to improve our article.

Second, I am attaching a letter responding to the comments and an improved version of the article.

Reviewer Response 1:

  1. According to the title of the article, the subject of this study was nurse assistants, but the role of the subjects cannot be found from the methods. The nursing system of each country is different. The author should add information related to the classification of nursing staff in their country and make it clear that these subjects are assistant nurses.

We clearly define who care nurses are in Chile.

"According to this classification, in Chile, assistance nurses work directly with people, providing care at different levels of care."

  1. The subject of the article is different from the keywords. For example, "occupational health" only appears in the keywords; "quality of work life" and "quality of life" have different meanings; "work process" does not appear, instead "working conditions".

These are the keywords because they are the most similar and include the concepts used in the research in their definition. Unfortunately, we cannot create our descriptors, so we use the available ones.

  1. In qualitative research, the sample was not representative because it was small and non-random.

In the qualitative design, the number of participants is not relevant to establishing the sample's representative, according to the definition based on Norman Densin(1). He declares that there was saturation when the object of study has been fully revealed.

  1. Denzin NK, Lincoln YS. The SAGE handbook of qualitative research. 5: e uppl. Los Angeles: SAGE; 2017.

Too many statements of personal characteristics in the article were unnecessary. The personal characteristics of the respondents in the abstract, discussion, and conclusion should be simplified.

Sociodemographic characteristics are now presented in a table, and there is only one paragraph in the discussion.

  1. There were 8 respondents in this article, but the authors only cited a few of the descriptions in the results, and did not make a generalization of the results of all the respondents. This method of presentation was not general and would cause bias. It is suggested that authors should describe the summary of all the respondents' answers in each paragraph, and then use individual respondent's statements as reinforcement.

A more significant number of verbatims were included in each subcategory in this new version. In the first version of the article, only one verbatim was included per subcategory.

  1. Many literatures were cited in the discussion, but no literature was compared with the results of this study. Therefore, it is not known whether the results of this study are valuable, nor can the results of this study inspire them.

The comparison of the results with the articles already cited is improved.

  1. The conclusion should state the important findings of the results of this study. It is not necessary to state the personal characteristics of the subjects, and there is no need to repeat the objective of the research.

Although the conclusion only followed the basic structure of three paragraphs: the first one informs if the objective was met, the second summarizes the main findings, and the third their contribution to public health. Its wording has been improved so as not to repeat the previously mentioned objective and results.

Reviewer 2 Report

  1. The introduction includes a wide range of information. Please clarify “what has already known” and “what you newly tried to expand” within the paragraphs.
  2. Please show the Themes and statements of the participants by using a table, not just the description.
  3. The study applies a purposive sampling approach. How did you apply the sampling ? For example, in some studies, researchers use a "directory" or some kind of database to track/find the respondents. Here, how did you track/find the respondents? In the current version, the sampling approach sounds more like a "convenience sampling approach". You need to clearly spell out how did you find the respondents for applying the purposive sampling approach.
  4. The issue(Quality of Work Life ) is not new. what is the contribution of your study?
  5. I suggest that you had better add a paragraph for limitations of this study at the end of the discussion. This study has more limitations that must be considered. Although the strategy of intentionally selecting participants is fully congruent with qualitative research, this might result in non-representative samples(only eight interviews).

Author Response

First of all, dear reviewer, I appreciate your time and comments to improve our article.

Second, I am attaching a letter responding to the comments and an improved version of the article.

Reviewer Response 2:

  1. The introduction includes a wide range of information. Please clarify "what has already known" and "what you newly tried to expand" within the paragraphs.

The last paragraph of the introduction clarifies what is already known and what this research intends to describe.

  1. Please show the Themes and statements of the participants by using a table, not just the description.

Since he is the only reviewer who has requested to make a table, it is preferred not to make it as this goes against the traditional presentation of data only in the narrative. In addition, the number of verbatims in each subcategory has been expanded.

  1. The study applies a purposive sampling approach. How did you apply the sampling ? For example, in some studies, researchers use a "directory" or some kind of database to track/find the respondents. Here, how did you track/find the respondents? In the current version, the sampling approach sounds more like a "convenience sampling approach". You need to clearly spell out how did you find the respondents for applying the purposive sampling approach.

As explained in data collection (lines 111 to 118): "The researchers contacted the head nurse of each institution to coordinate the interviews, inviting all nurses in each workplace to participate. According to time availability, individual interviews were agreed upon, developed by the researcher without a prior relationship with participants."

The issue (Quality of Work Life ) is not new. what is the contribution of your study?

This study provides the quality of work life of the nurse dedicated to the assistance role both at the primary and tertiary level of care in Chile. The object of study that to date has been insufficiently revealed.

  1. I suggest that you had better add a paragraph for limitations of this study at the end of the discussion. This study has more limitations that must be considered. Although the strategy of intentionally selecting participants is fully congruent with qualitative research, this might result in non-representative samples (only eight interviews).

The limitation paragraph at the end of the discussions was improved.

Reviewer 3 Report

Dear Authors, 

I am sending my comments.

Introduction - ok

Methodology:

a) Inclusion criteria: workday 22-33-44 hrs - what does it mean? I do not understand hourly intervals, why such?

b) It is not clear from the instrument description whether an original tool was developed or whether it is a combination of existing QWL survey tools. The authors wrote that the tool was developed after a literature review. This needs to be described in detail and if some of the questions are from a tool that already exists, it needs to be cited. 

c) Descriptive statistics were used - i.e., what was calculated and how?

Results:

a) Sociodemographic data is worth presenting in tabular form - it is more readable. Please correct. 

Disscusion:

The discussion is unreadable. The discussion should be a structured comparison of the results obtained against other convergent work. It needs structural and qualitative reconstruction. 

Conclusion

Some results that should be in the "Results" section have been moved to the applications.
Needs improvement. 

Author Response

First of all, dear reviewer, I appreciate your time and comments to improve our article.

Second, I am attaching a letter responding to the comments and an improved version of the article.

Reviewer Response 3:

Methodology:

  1. a) Inclusion criteria: workday 22-33-44 hrs - what does it mean? I do not understand hourly intervals, why such?

According to the administrative statute that governs public institutions in Chile, employment contracts can be 22 and 33 hours (part-time) and 44 full-time hours. Therefore, the respective paragraph in the text has been improved.

  1. b) It is not clear from the instrument description whether an original tool was developed or whether it is a combination of existing QWL survey tools. The authors wrote that the tool was developed after a literature review. This needs to be described in detail and if some of the questions are from a tool that already exists, it needs to be cited.

The instrument is an original interview guide created by the authors. The paragraph has been improved.

  1. c) Descriptive statistics were used - i.e., what was calculated and how?

Absolute frequencies and cumulative frequencies were used. In addition, the respective paragraph has been improved.

Results:

  1. a) Sociodemographic data is worth presenting in tabular form - it is more readable. Please correct.

The sociodemographic data have been placed in a table.

Disscusion:

The discussion is unreadable. The discussion should be a structured comparison of the results obtained against other convergent work. It needs structural and qualitative reconstruction.

The discussion is rewritten to improve its readability and quality. Improving the comparison of the results with the articles already cited.

Conclusion

Some results that should be in the "Results" section have been moved to the applications. Needs improvement.

Although the conclusion only followed the basic structure of three paragraphs: the first one informs if the objective was met, the second summarizes the main findings, and the third their contribution to public health. Its wording has been improved so as not to repeat the previously mentioned objective and results.

Round 2

Reviewer 1 Report

Each topic should first describe a summary result, and then use the individual respondent's statements to support it

Author Response

Thanks for the new review of the article and the new comments. Regarding what is requested, each subcategory is defined at the beginning of this, complementing it with the respective narratives.

Reviewer 2 Report

The authors responded my comments and revised the discussion section and limitation. This revise manuscript became better understanding quality of work life and work process of assistance nurses in Chile.  So, I think this revise manuscript became suitable for accepting the journal now.

Author Response

Thanks for the new review of the article.

The results have been improved at the request of the other reviewer.